# Bandwidth Row Ratio Configuration Affect Interspecific Effects and Land Productivity in Maize–Soybean Intercropping System

Liang Feng [1,2,3,†], Wenting Yang [1,†], Haiying Tang [4], Guoqin Huang [1] and Shubin Wang [1,*]

1   Key Laboratory of Crop Physiology, Ecology and Genetic Breeding (Jiangxi Agricultural University), Ministry of Education, Nanchang 330045, China
2   College of Agronomy, Sichuan Agricultural University, Chengdu 611130, China
3   Sichuan Engineering Research Center for Crop Strip Intercropping System, Key Laboratory of Crop Ecophysiology and Farming System in Southwest China (Ministry of Agriculture), Chengdu 611130, China
4   College of Agriculture and Biotechnology, Hunan University of Humanities, Science and Technology, Loudi 417000, China
\*   Correspondence: shubinwjxau@126.com
†   These authors contributed equally to this work.

**Abstract:** Intercropping plays an indispensable role in sustainable agriculture. The response of bandwidth row ratio configuration to crop interspecific relationships and land productivity in the maize–soybean intercropping system (MSI) is still unclear. A 2-year field experiment was conducted with sole maize (SM) and sole soybean (SS), two different bandwidths (2.4 m (B1), 2.8 m (B2)), two different maize and soybean row ratios (2:3 (R1), and 2:4 (R2)) for MSI. The results showed that intercropping had advantages for land productivity compared with sole planting. Intercropping cropping had significant differences on crop yield under different intercropping treatments. The 2-yr average land equivalent ratio (LER, 1.59) and group yield under the intercropping patterns of B1R2 were significantly higher than other intercropping treatments ($p < 0.05$). With a bandwidth of 2.4 m and planting four rows of intercropped soybean, the total LER and group yield increased by 7.57% and 10.42%, respectively, compared to planting three rows of soybean. Intercropped maize was the dominant species and also had a higher nutrient aggressivity than intercropped soybean. The complementarity effect was higher than the select effect in the MSI system, and intercropping advantage was mainly derived from the complementarity effect, which was significantly correlated with intercropped maize yield. Nitrogen and phosphorus nutrient aggressivity in intercropped maize showed significant correlations with group yield and intercropped maize yield. In conclusion, bandwidth 2.4 m, row ratio 2:4 was a reasonable planting pattern because of its superior land productivity, crop nutrients uptake advantage, and harmonious interspecific relationship, which could provide a reference for MSI promotion and application research.

**Keywords:** intercropping; nutrient accumulation; interspecific relationship; aggressivity; land productivity

## 1. Introduction

The human population is expected to reach 10 billion by 2050 [1]. However, crop production appears to be plateauing and even declining due to global warming and decreasing arable land. Improving agricultural productivity and sustainability is vital to the world [2]. Maize (*Zea mays* L.) and soybean (*Glycine max* L.) are parts of the four major grain crops (rice (*Oryza sativa*), wheat (*Triticum aestivum*), maize, and soybean), which play an essential role in stabilizing food security [3]. Intercropping is a food-production system of cultivating two or more crops are grown simultaneously on the same land [4]. Maize–soybean intercropping (MSI) is one of the typical representatives of gramineae-legume intercropping systems [5]. MSI could improve crop yield and income [6], increase land productivity and resource (light, heat, water, and nutrients) use efficiency [7,8], reduce pests and diseases [9], and enhance biodiversity [10,11]. MSI could alleviate the contradiction

between population growth and arable land reduction, which has been applied in many countries around the world [3].

In intercropping systems, interspecific competition and promotion effects co-exist between various crop species, and efficient use of resources by crops could be achieved by coordinating interspecific relationships [12,13]. During the intercrops' co-growth period, plants usually compete for light, heat, water, and nutrients in a limited space due to ecological niche differences. Therefore, the construction of a balanced and stabilized intercropping system is essential to boosting crop yield. In MSI systems, taller maize will preferentially capture sunlight due to the higher niche, which is conducive to increased biomass and grain yield [14]. In contrast, the interception and utilization of sunlight by dwarf soybean will be diminished due to the shade effect [15]. Therefore, one of the most crucial measures to enhance yield is to optimize the planting strategy of the intercropping system [16]. Interspecific relationships occurred not only at the spatial scale but also at the horizontal scale; the interactions mainly occurred at the intersection interface between strips, and the intercropping effect would gradually decrease with the widening of the strips [17]. Different MSI patterns with bandwidth row ratio configurations not only affect interspecific relationships but also nutrients competition, and relationships between nutrient aggressivity of intercropped crops and land productivity of intercropping systems need to be further investigated.

Select effect (SE) and complementarity effect (CE) are commonly used to explain the interrelationship between biodiversity and system productivity [18]. SE indicated that more diverse mixtures have a higher probability of containing high-yielding species [19], and CE refers to spatial-temporal ecological niche separation and interspecific promotion, which could acquire more resources for the crops and improve productivity [20,21]. In intercropping systems, nutrient addition (N or P) may affect crop SE and CE by influencing the nutrient uptake of the species [18]. SE and CE may also be heterogeneous between species due to nutrient uptake and utilization differences between intercropped crops [22,23]. However, the different planting patterns possess different ecological niches and nutrient uptake capacities, and it is not clear how this planting pattern affects crop SE and CE. The study of SE and CE could help us to further understand the mechanisms of intercrop dominance when applied to intercropping systems.

This study was conducted as a 2-year field experiment using different configurations of bandwidth (2.4 m, 2.8 m) and maize–soybean row ratios (2:3, 2:4). We hypothesize that MSI with different bandwidth row ratio configurations could improve land productivity and coordinate interspecific competition to achieve high yield and high efficiency cultivation by optimizing field layout. The main objectives were: (i) quantifying the yield and nutrient absorption advantage of MSI compared to sole planting; (ii) analyzing the relative importance of SE and CE under different bandwidth row ratio configurations; and (iii) clarifying the relationships among nutrient aggressivity, CE, SE, intercrop advantage, and land productivity.

## 2. Materials and Methods

### 2.1. Site Description

Field experiments were conducted in 2018 and 2019 at the Jiangxi Institute of Red Soil (28°15′ N, 116°20′ E) in Jinxian County, Jiangxi Province, China. The experimental site has a central subtropical monsoon climate, with a mean annual air temperature of 17.7~18.5 °C and annual precipitation of 1537 mm. The base fertility of red soil is presented in Table 1, and the mean precipitation and temperature during the experimental years are presented in Figure 1.

**Table 1.** Agro-chemical properties of test red soils.

| Years | pH | OM g kg$^{-1}$ | AN mg kg$^{-1}$ | AP mg kg$^{-1}$ | AK mg kg$^{-1}$ |
|---|---|---|---|---|---|
| 2018 | 5.74 | 19.79 | 105.00 | 14.22 | 257.01 |
| 2019 | 5.72 | 19.7 | 102.76 | 12.8 | 174.08 |

Abbreviations: OM = organic matter; AN = available nitrogen; AP = available phosphorus; AK = available potassium.

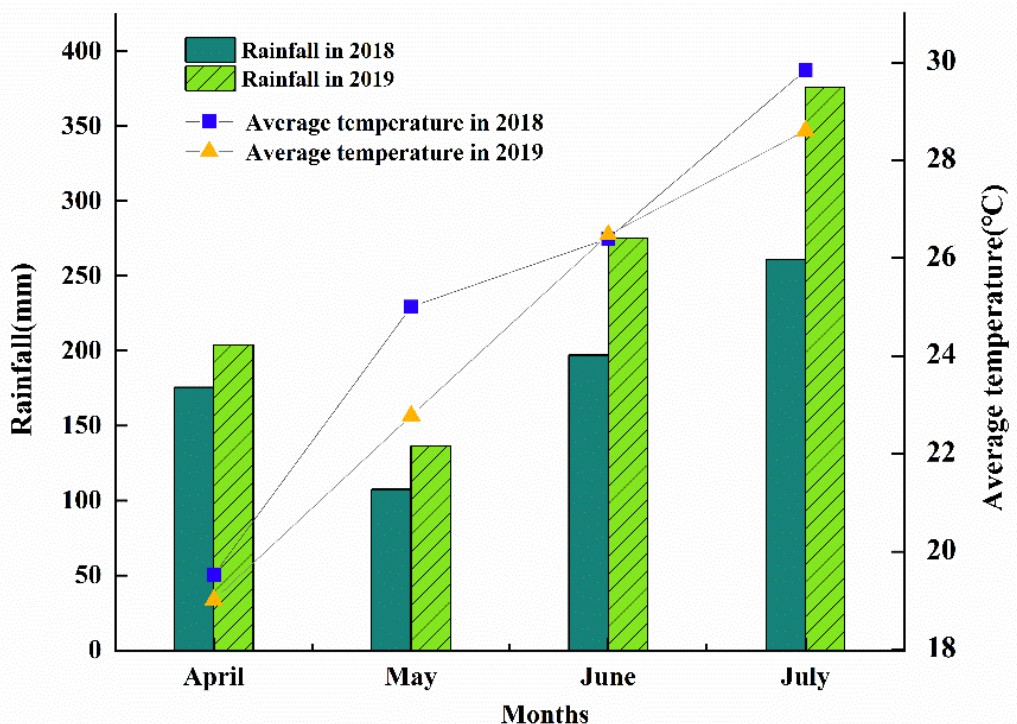

**Figure 1.** Monthly precipitation and average air temperature during maize and soybean growing period in two years.

*2.2. Experimental Design and Field Management*

The compact maize variety (Jixiang-1) and the local high-yielding soybean variety (Handou-1) were selected for field experiments. The experiment was a randomized complete block design with three replications, six different treatments, and a total of 18 plots, i.e., sole maize (SM), sole soybean (SS), bandwidth 2.4 m, two-rows maize intercropped with three-rows soybean (B1R1), bandwidth 2.4 m, two-rows maize intercropped with four-rows soybean (B1R2), bandwidth 2.8 m, two-rows maize intercropped with three-rows soybean (B2R1), bandwidth 2.8 m, two-rows maize intercropped with four-rows soybean (B2R2). Each plot was planted with two belts of crops, B1R1 and B1R2 experimental plots were 24 m$^2$ (4.8 m wide × 5 m long), B2R1 and B2R2 experimental plots were 28 m$^2$ (5.6 m wide × 5 m long), the areas of sole maize and sole soybean were 17.5 m$^2$ and 12.5 m$^2$, respectively.

The bandwidth row ratio configuration was presented in Figure 2 and Table 2. The plant populations of maize and soybean for sole and intercrops were all 60 K ha$^{-1}$ and 150 K ha$^{-1}$. A total of 270 kg ha$^{-1}$ of pure N was applied to maize during the whole period, in basal fertilizer: jointing fertilizer: panicle fertilizer = 3:2:5, 72 kg pure P, and 90 kg pure K prior to sowing intercropped and sole maize, and 34.5 kg pure N, 72 kg pure P, and 36 kg pure K for soybean basal fertilizers. Maize was sown on 9 April 2018 and 8 April 2019 and harvested 22 July 2018 and 30 July 2019. Soybean was sown on 9 April 2018 and 8 April 2019 and harvested 9 July 2018 and 11 July 2019. All agronomic practices, i.e., sowing, harvesting, and weeding, were performed manually according to local farmers' practices in this region.

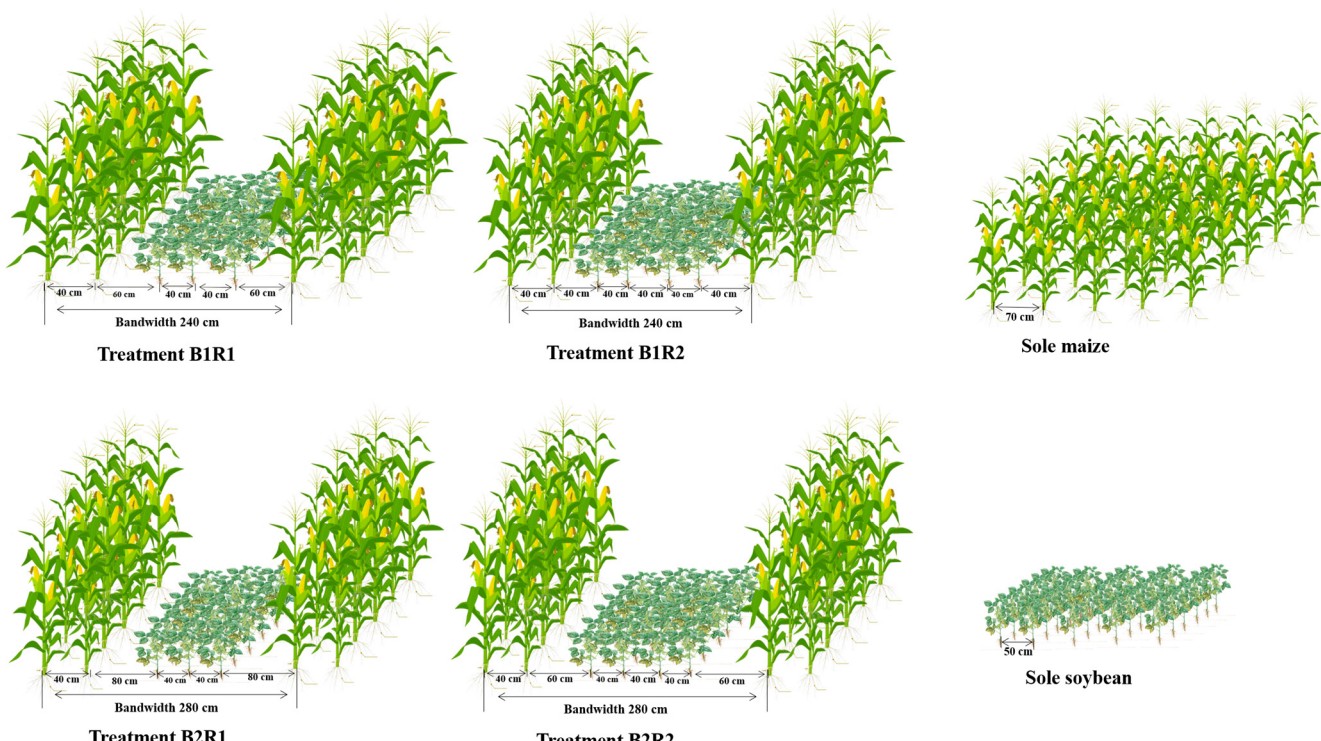

**Figure 2.** Schematic illustrations of different bandwidth and row ratio configurations. Four different intercrop treatments (B1R1, B1R2, B2R1, and B2R2) and two different sole treatments (sole maize/sole soybean). Two belt crops were planted in one plot, each belt consisted of two rows maize and three or four rows soybean.

**Table 2.** Field configuration test design of different bandwidth and row ratio. (Unit: cm).

| Treatments | Bandwidth | Maize-Soybean Row Ratio | Seed Spacing (Maize/Soybean) | Maize-Soybean Row Distance |
|---|---|---|---|---|
| B1R1 | 240 | 2:3 | 13.8/8.3 | 60 |
| B1R2 | 240 | 2:4 | 13.8/11 | 40 |
| B2R1 | 280 | 2:3 | 11.9/7.1 | 80 |
| B2R2 | 280 | 2:4 | 11.9/9.5 | 60 |
| SM | — | — | 23.8 | — |
| SS | — | — | 13.3 | — |

B1R1—bandwidth 2.4 m, two rows maize with three rows soybean. B1R2—bandwidth 2.4 m, two rows maize with four rows soybean. B2R1—bandwidth 2.8 m, two rows maize with three rows soybean. B2R2—bandwidth 2.8 m, two rows maize with four rows soybean. SM—sole maize, SS—sole soybean.

### 2.3. Data Collection and Measurements

2.3.1. Grain Yield and N, P and K Content

Each experimental plot, consisting of two belts of maize and soybean, was roughly divided into two sections. One belt was used for analyzing N, P, and K element content; in this belt, three consecutive maize plants and six consecutive soybean plants were collected at the mature stage, whole plants were cut at ground level using pruning scissors. After sampling, different crop organs, including the straw (leaf + stem) and grain, were separated as soon as possible, and then the plant tissues were oven-dried at 105 °C for 1 h to destroy the tissues and dried at 80 °C to a constant weight before weighing. Samples of oven-dried crop were digested in a mixture with $H_2SO_4$ and $H_2O_2$, and N, P, and K contents of samples were measured by micro-Kjeldahl procedure, vanadomolybdate method, and flame photometry, respectively [24]. Harvesting another belt of intercropped crops of maize and soybeans, threshed by hand and dried under the sun for at least 10 days to achieve a constant weight, then weighed to the grain yield and changed into kg ha$^{-1}$.

2.3.2. Nutrient Accumulation

$$\text{Nitrogen accumulation (NA)} = \text{weight of dry matter (kg ha}^{-1}) \times \text{N (\%)}/100. \quad (1)$$

Crop phosphorus accumulation (PA) and potassium accumulation (KA) were calculated in accordance with NA.

2.3.3. Intercropping Advantage

(1)　Land equivalent ratio (LER): as a measure of yield advantage [25].

$$\text{LER}_M = \text{Yim/Ysm}, \text{LER}_S = \text{Yis/Yss}, \text{LER} = \text{LER}_M + \text{LER}_S \quad (2)$$

where $\text{LER}_M$ and $\text{LER}_S$ are the partial land equivalent ratios of intercropped maize and soybean, respectively. Yim and Yis are the intercropped maize and intercropped soybean grain yields, respectively. Ysm and Yss are the sole maize and sole soybean grain yields, respectively. When LER > 1, it shows intercropping advantage, and when LER < 1, it shows intercropping disadvantage.

(2)　Nitrogen equivalent ratio (NER): As a measure of nitrogen uptake advantage in crops.

$$\text{NER}_M = \text{Nim/Nsm}, \text{NER}_S = \text{Nis/Nss}, \text{NER} = \text{NER}_M + \text{NER}_S \quad (3)$$

where $\text{NER}_M$ and $\text{NER}_S$ are the partial nitrogen equivalent ratios of maize and soybean, respectively. Nim and Nis represent nitrogen accumulation per unit area in aboveground intercropped maize and intercropped soybean, respectively. Nsm and Nss represent nitrogen accumulation per unit area in aboveground sole maize and sole soybean, respectively. When NER > 1, it shows intercropping advantage, and when NER < 1, it shows intercropping disadvantage. The phosphorus equivalent ratio (PER) and potassium equivalent ratio (KER) are calculated in the same way as the NER.

(3)　Net effect (NE), complementarity effect (CE), and select effect (SE): The NE includes two components: SE and CE [18].

$$\text{NE} = (Y_{is} + Y_{im}) - (Y_{ss} \times P_s + Y_{sm} \times P_m) \quad (4)$$

$$\text{CE} = \left(\left(\frac{Y_{is}}{Y_{ss}} + \frac{Y_{im}}{Y_{sm}}\right) - 1\right) \times \left(\frac{Y_{ss} + Y_{sm}}{2}\right) \quad (5)$$

$$\text{SE} = \text{NE} - \text{CE} \quad (6)$$

NE > 0, indicating a positive effect between diversity planting and intercropping system productivity, NE < 0, indicating a negative effect. SE > 0, indicates that high yielding species will be increased in the intercropping system, SE < 0, low yielding species will be increased in the intercropping system. Pm and Ps represent the percentage of plot area occupied by intercropped maize and intercropped soybean, respectively. In our study, the plots were occupied by maize as Pm = (maize-maize row spacing + maize-soybean row spacing)/bandwidth, for example, the area occupied by maize in plot B1R1 was Pm = (40 cm + 60 cm)/240 cm = 0.42, the area occupied by maize in the other plots was 0.33, 0.43, 0.36, and the area occupied by soybeans in the plots was 0.58, 0.67, 0.57, and 0.64 [7]. Yim and Yis are the intercropped maize and intercropped soybean yields, respectively. Ysm and Yss are the sole maize and sole soybean yields, respectively.

(4)　Nutrient Aggressivity (NA): indicates the relative nitrogen uptake of intercrop A compared to the advantage of nitrogen uptake of intercrop B.

$$\text{NA}_M = \text{Nim/Nsm} \times \text{Pm} - \text{Nis/Nss} \times \text{Ps} \quad (7)$$

$$\text{NA}_S = \text{Nis/Nss} \times \text{Ps} - \text{Nim/Nsm} \times \text{Pm} \quad (8)$$

$\text{NA}_M$ and $\text{NA}_S$ are represented as intercropped maize nitrogen aggressivity and intercropped soybean nitrogen aggressivity, respectively. $\text{NA}_M = \text{NA}_S$, indicated that maize and soybean had the same competitive advantage for nitrogen uptake; $\text{NA}_M > \text{NA}_S$ means that intercropped maize has a stronger nitrogen uptake advantage than intercropped soybean; $\text{NA}_M < \text{NA}_S$ means that intercropped maize has less advantage in nitrogen uptake

than intercropped soybean. The phosphorus aggressivity (PA) and potassium aggressivity (KA) are calculated in the same way as the NA.

### 2.4. Statistical Analysis

Data were analyzed using Excel 2019 (Microsoft, Redmond, DC, USA) [7], the least significant difference (LSD) and Duncan method were used for post-hoc multiple comparisons and difference significance test. The vegan package using the R (4.2.1 version) was used to analyze the mantel correlation between each factor of intercrop advantage and crop yield.

### 3. Results

#### 3.1. Effect of Different MSI Patterns on Crops Nutrients Accumulation

On average, B1R2 pattern intercropped maize showed the highest N accumulation, compared to B1R1 intercropped maize, where N accumulation increased by 2.73% (Figure 3A). B1R2 patterns had a higher maize P accumulation of 15.43 kg ha$^{-1}$, which had no significant differences from the other intercropping treatments ($p > 0.05$) (Figure 3B). B1R2 intercropped soybean had the highest N accumulation of 41.97 kg ha$^{-1}$ (Figure 3A). B2R2 intercropped soybean had the highest P accumulation, which had significant differences from B1R1 ($p < 0.05$, Figure 3B). B1R2 pattern intercropped soybean K accumulation had significant differences from other intercropping treatments, and compared to B1R1, intercropped soybean K accumulation increased by 29.35% ($p < 0.05$, Figure 3C).

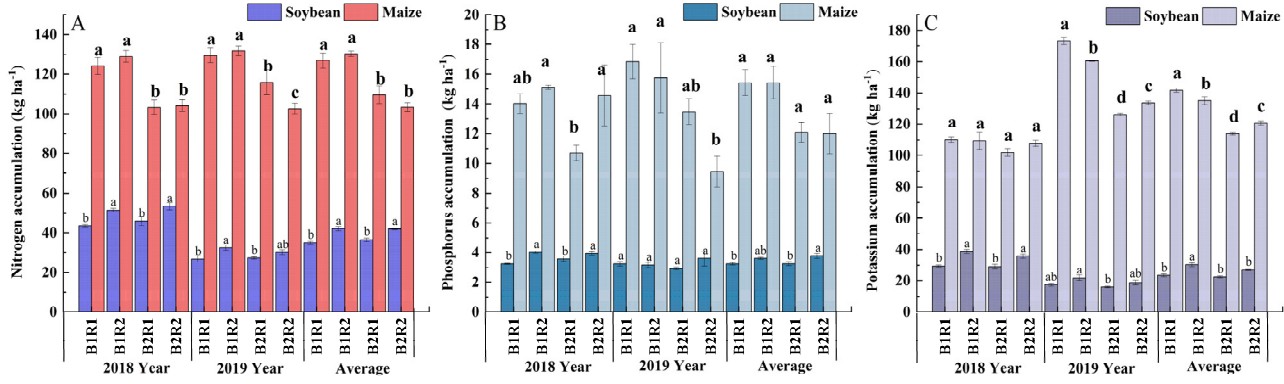

**Figure 3.** Nutrient accumulation of maize and soybean in sole and intercropping systems in the 2 yr. (**A**–**C**) represent the N, P, and K accumulation among different intercropping treatments, respectively. Different lowercase letters above the bars indicate significant differences ($p < 0.05$). Values are means ± standard error.

#### 3.2. Effect of Different MSI on Crop Intercrop Advantage

The intercropping system had the yield advantage and nutrients uptake advantage compared to monoculture, with all LER and nutrients equivalent ratios greater than one (Figure 4). The B1R2 patterns LER$_M$ and LER showed a superior advantage and were significantly different from other intercropping treatments ($p < 0.05$, Figure 4C). The bandwidth was 2.4 m or 2.8 m, the row ratio increased from 2:3 to 2:4, and the LER$_M$ and LER$_S$ were all increased. For a certain row ratio, when expanding the bandwidth, both LER$_M$ and LER were all decreased.

In 2018, B1R2 patterns NER, PER, and KER were highest, compared with B1R1 patterns NER, PER and KER increased by 9.22%, 14.5%, and 12.5%, respectively (Figure 4D). In 2019, B1R2 patterns NER and KER were both the highest at 2.01 and 1.93 (Figure 4E). With bandwidth of 2.4m or 2.8m, the row ratio increased from 2:3 to 2:4, which was beneficial to improving the KER. With a maize–soybean row ratio of 2:3 or 2:4, the bandwidth increased from 2.4 m to 2.8 m, and NER, PER, and KER were decreased in all intercropping treatments.

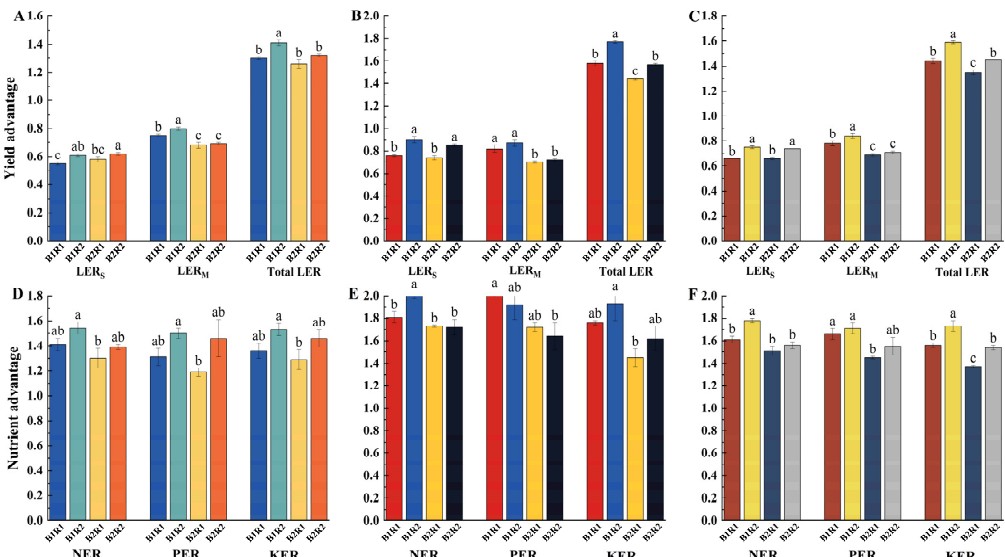

**Figure 4.** (**A**,**B**) represent the land equivalent ratios between 2018 and 2019, respectively, and (**C**) represents the average 2 yr land equivalent ratio. (**D**,**E**) represent the nutrients equivalent ratio between 2018 and 2019, respectively, and (**F**) represents the average 2yr nutrients equivalent ratio. Lowercase letters above the bars indicate significant differences ($p < 0.05$). Values are means $\pm$ standard error.

### 3.3. Effects of Different MSI on Interspecific Relationships of Crops

In 2018, B1R2 patterns showed superior $NA_M$ and $PA_M$ of 1.68 and 1.28, respectively (Figure 5A,B). In 2019, there were no significant differences in $NA_M$ and $PA_S$ in intercropping treatments ($p > 0.05$, Figure 5D,E). $NA_M$, $PA_M$, and $KA_M$ were greater than zero, but the $NA_S$, $PA_S$, and $KA_S$ were lower than zero. B1R2 patterns could increase $NA_M$, $PA_M$, and $KA_M$. When the maize–soybean row ratio was 2:3, the bandwidth was increased from 2.4 m to 2.8 m, which was beneficial to increase $NA_S$, $PA_S$, and $KA_S$.

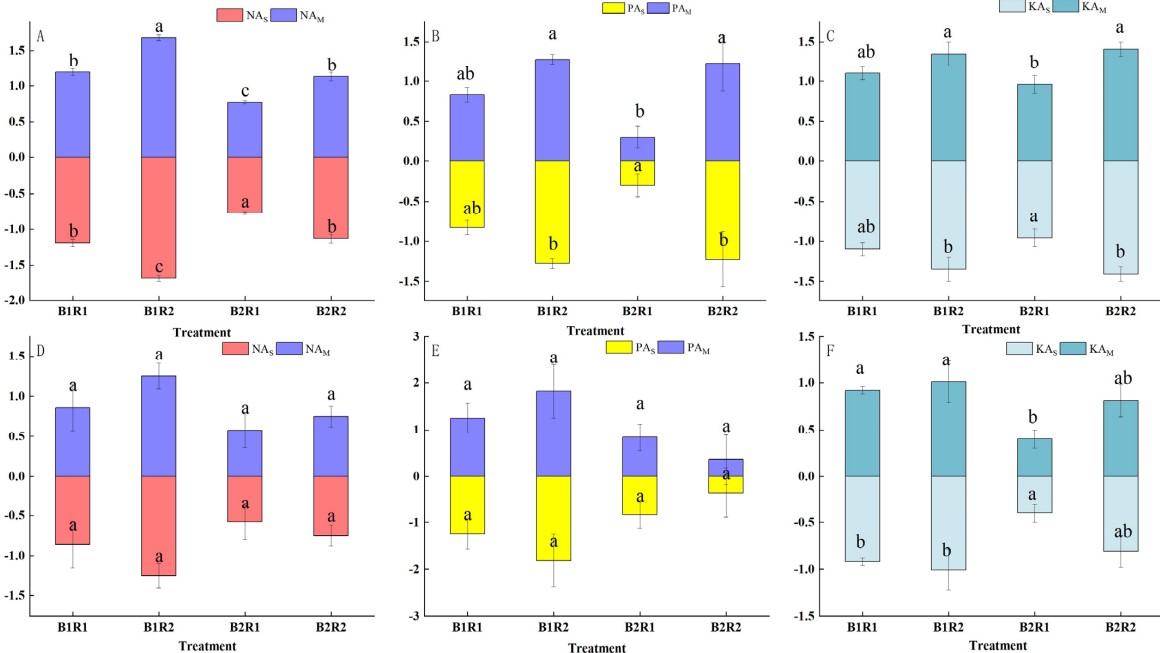

**Figure 5.** Nutrient aggressivity of maize–soybean intercropping systems. (**A**–**C**) represent intercropped maize and intercropped soybean N, P, and K aggressivities in 2018, respectively. (**D**–**F**) represent intercropped maize and intercropped soybean N, P, and K aggressivities in 2019, respectively. lowercase letters above the bars indicate significant differences ($p < 0.05$). Values are means $\pm$ standard error.

### 3.4. Effects of Different MSI Patterns for Crop Yield

In 2018, B1R2 patterns showed the highest yield of intercropped maize, which was a significant difference compared to other intercropping treatments (Table 3, $p < 0.05$). In 2019, the solo maize crop yield increased by 14.54% compared to B1R2. The B1R2 pattern group yield had significant differences from other intercropping treatments ($p < 0.05$). When the bandwidth was 2.4 m or 2.8 m, the row ratio increased, which was beneficial to increasing the mean yield of intercropped crops. However, when the row ratio was 2:3 or 2:4, expanding the bandwidth distance, which was not conducive to increasing crop yield.

**Table 3.** Effect of bandwidth row ratio configuration on crop yield.

| Year | Treatments | Maize | Soybean | Group |
|---|---|---|---|---|
| | | kg ha$^{-1}$ | | |
| | B1R1 | 4797.00 c | 673.20 d | 5470.20 b |
| | B1R2 | 5146.50 b | 743.07 bc | 5889.57 a |
| 2018 | B2R1 | 4355.73 d | 705.60 cd | 5061.33 c |
| | B2R2 | 4450.00 d | 759.53 b | 5209.53 c |
| | Sole | 6404.75 a | 1222.00 a | |
| | B1R1 | 6550.00 b | 627.85 c | 7177.85 b |
| | B1R2 | 6977.78 b | 736.41 b | 7714.19 a |
| 2019 | B2R1 | 5595.24 c | 611.90 c | 6207.14 c |
| | B2R2 | 5780.95 c | 700.27 b | 6481.22 c |
| | Sole | 7992.38 a | 823.11 a | |

Note: group yields are the sum of the yields produced by the two intercropped crops. Within the same column in the same year, different lowercase letters indicate significant difference ($p < 0.05$).

### 3.5. Effect of Different MSI Patterns on NE, SE and CE

The average NE, SE, and CE of the intercropping system were all greater than zero (Figure 6). The NE, SE, and CE of the B1R2 pattern were 3741.24 kg ha$^{-1}$, 1261.97 kg ha$^{-1}$, and 2479.27 kg ha$^{-1}$, respectively, which had significant differences compared to other intercropping treatments ($p < 0.05$). With the same bandwidth and increasing row ratio, the NE, SE, and CE showed an increasing trend of intercropping. With the same row ratio and increased bandwidth, the NE, SE, and CE showed a decreasing trend for intercropping.

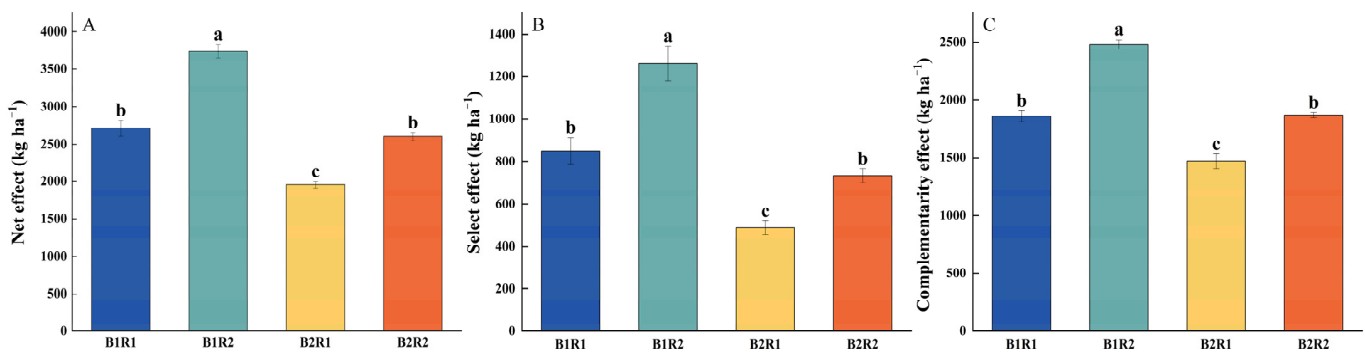

**Figure 6.** (**A–C**) indicate the net effect, select effect, and complementarity effect, respectively. Differently colored bars indicate different intercrop treatments. Lowercase letters above the bars indicate significant differences ($p < 0.05$). Values are means ± standard error.

### 3.6. Correlation Analysis of Intercropped Crops on Nutrient Aggressivity and Land Productivity

According to the Mantel test analysis, SY had a highly significant positive correlation with LER$_S$ ($p < 0.001$). MY had a highly significant positive correlation with NA$_M$, LER$_M$, NA$_S$, PA$_S$, NE, CE, SE, and total LER, respectively ($p < 0.001$). GY had a highly significant positive correlation with NA$_M$, LER$_M$, NA$_S$, PA$_S$, NE, CE, and SE, respectively ($p < 0.001$).

NA$_M$ had a highly significant positive correlation with total LER, CE, NE, SE, and LER$_M$, respectively ($p < 0.001$). NA$_S$ had a highly significant negative correlation with

total LER, CE, NE, SE, and $LER_M$, respectively ($p < 0.001$). NE, CE, and SE had a highly significant positive correlation with each other ($p < 0.001$). NE, CE, and SE were highly significantly positively correlated with $LER_M$ ($p < 0.001$) and highly significant negative correlations with $NA_S$ ($p < 0.001$).

## 4. Discussion

### 4.1. Different Planting Patterns in Response to Land Productivity

The intercropping system has an obvious promotion effect on the growth and crop yield, which was also essential to enhance the land productivity [26]. One of the measures for constructing a reasonable spatial layout is to regulate the planting structure of inter-cropping crops [27], which was intended to reduce the competitive effect and achieve the interspecific reciprocity effect of the groups so as to improve intercropping advantage and land productivity [10]. In the present study, the LER, NER, PER, and KER were all greater than one under all intercropping systems (Figure 4), which indicated that intercropping has the superior yield and nutrient uptake advantage [13]. The 2.4 m bandwidth, 2:4 row ratio pattern had a greater average LER (1.59) than others in the 2-yr period, which had greater land productivity. It was comprised of previous research results [28].

Under a bandwidth of 2.4 m, a row ratio of 2:3 patterns, and the addition of one row soybean, the average $LER_S$, $LER_M$, and LER were all increased. The reasons could be that (1) the interspecific distance was reduced and belowground interactions were intensified [29], and in addition, the root system of leguminous crops secretes chemical substances, etc., which were beneficial to improve the root activity and root distribution of intercropped maize [30] and promote the uptake and utilization of soil nutrients by intercropped maize [31,32]. (2) intercropped maize could uptake N transferred from the intercropped soybean roots, and superior N provided better nutrient conditions for intercropped maize growth [33,34]. In addition, the maize–soybean row ratio was 2:3 or 2:4, so expanding bandwidth from 2.4 m to 2.8 m would not increase land productivity. The reasons could be that (1) plant-to-plant spacing decreased, the more individuals per unit area leaded to increased intraspecific competition and breakdown of stable spatial ecological niches, resulting in reduced leaf area index due to mutual shading among individual crops [35], and affected crop growth due to insufficient soil nutrients supplied for individual plants [36]. (2) The increased evaporation of soil water because of the wider interspecific distance, which was not conducive to crop water uptake, resulting in a reduced yield advantage of intercropped maize [25].

The LER in 2019 was higher than that in 2018, which might be due to the average air temperature in 2018 was higher than that in 2019 during the whole growing period (Figure 1). On the one hand, average air temperature would affect soil moisture and photorespiration rate [37]. Meanwhile, the crop growing season had better rainfall in 2019 (Figure 1), and the superior water supply promoted the synthesis of photosynthetic products and the accumulation of nutrients in sink [8] and could increase maize yield. In addition, experiment errors during sample collection and production process, pests, and diseases also affected LER. The yield of soybean was higher, but the yield of maize was lower in all intercropping treatments in 2018 compared to 2019. The reason could be longer precipitation affected photosynthesis in intercropped soybean in 2019, while causing disease increased in intercropped soybean shoot and soybean grain rot, resulting in lower yields [38]. On the contrary, maize as the high-straw crop, abundant precipitation favors crop growth and yield increase [39].

### 4.2. Response of Different Planting Patterns to the Competitive of N, P and K in Crops

Different bandwidth row ratio configurations affect crops' nutrients accumulation (Figure 3), and intercropped maize had higher nutrient accumulation and nutrient aggressivity than intercropped soybean. On the one hand, intercropped maize was the competitively dominant species; on the other hand, maize as gramineous crops had the greater nutrient uptake capacity and higher biomass yield [29]. In addition, the co-growth

of maize and soybean for a long time makes the competition for resources between intercropping species continuous, while maize as a competitively dominant species would limit the growth of soybean, especially the shade effects [15].

Intercropped maize has higher nutrients aggressivity and nutrients accumulation under a bandwidth of 2.4 m and a row ratio in a 2:4 pattern, which, because intercropped maize could obtain abundant N from the rhizosphere soil and neighboring soybean [40], then the N concentration might be reduced in soybean strip soil, promoting $N_2$ fixation by soybean root nodules, and alleviating the soybean N inhibitory effect [41], which facilitated the complementarity of nutrients utilization and balanced interspecific relationship to obtain higher group yield between gramineae and legumes [9].

When the bandwidth was 2.4 m and one row of soybean was added, it could increase the nutrient aggressivity of intercropped maize because (1) it may improve the belowground soil nutrient spatial ecological niche due to reduced interspecific distances, resulting in maize roots with increased horizontal growth capacity and thus improved intercrop maize aggressivity due to greater nutrient acquisition capacity [42]. When the maize–soybean row ratio was 2:3 or 2:4 and the bandwidth increased from 2.4 m to 2.8 m, it could improve the nutrient aggressivity of intercropped soybean; this may be due to expanding the interspecific distance, alleviating the shading effect of the high-straw maize on dwarf soybeans, and facilitating soybean interception of more photosynthetic active radiation and increasing nutrients and dry matter accumulation [43].

### 4.3. Response of Interspecific Effect to Crop Yield

In this study, crop yield showed a highly significant positive correlation with $PA_M$ and $NA_M$ (Figure 7), indicating that intercropped maize affected crop yield under different intercropping patterns mainly through the competitive ability of N and P, probably because legume crop root secretions could activate inter-root soil inorganic P and promote crop uptake of P and increase yield [4,14]. In addition, intercropping could improve soil texture and promote crop nutrients absorption [44], legume nodules where atmospheric $N_2$ was converted into ammonium ($NH_4^+$), which sustained growth of legumes and gramineaes [45], and application of chemical fertilizer has provided the essential nutrients for crop growth [46].

In this research, the co-effects of complementarity and select effect affected the productivity advantage of MSI. The $LER_M$ had a significant positive correlation with crop yield ($p < 0.001$), indicating that intercropping system land productivity was improved mainly due to the increased yield of intercropped maize [47], while intercropped maize yield had a significant positive correlation with complementarity and select effect ($p < 0.001$), indicating that the higher the complementarity effect and select effect, the greater the effect of increased maize yield. Complementarity effects were higher than select effects; in addition, intercropped soybean yields also had a significant positive correlation with the compensatory effect ($p < 0.01$), indicating that the intercropping advantage was mainly derived from complementarity effect [27], which was due to the competitive strength and broad-row light transmission of intercropped maize, which further strengthened after intercropped soybean harvest and improved the spatial ecological niche of the crops [48]. The complementarity effect, select effect, and net effect were the highest under the bandwidth of 2.4 m and row ratio of 2:4 pattern, which had obvious intercropping advantages, indicating that this pattern had a reasonable interspecific allocation strategy.

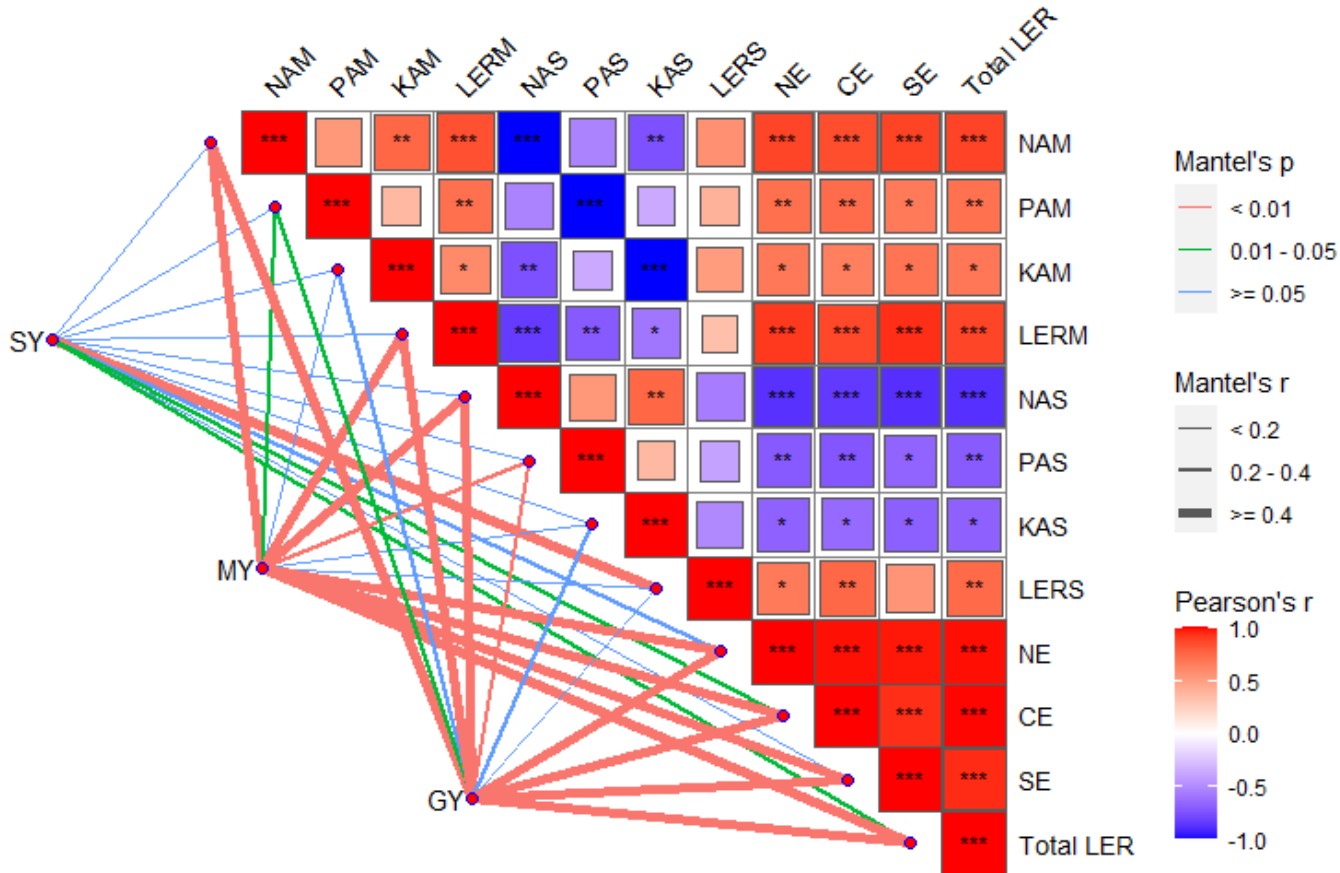

**Figure 7.** The size and color of the blocks in the figure indicate the strength of the positive and negative correlations among various factors. $NA_M$, $PA_M$, $KA_M$, and $LER_M$ represent N, P, and K nutrient aggressivity and partial land equivalent ratio for intercropped maize, respectively. $NA_S$, $PA_S$, $KA_S$, and $LER_S$ represent N, P, and K nutrient aggressivity and partial land equivalent ratio for intercropped soybean, respectively. NE, CE, and SE represent net, complementarity, and select effect, respectively. SY, MY, and GY represent intercropped soybean yield, intercropped maize yield, and group yield, respectively. The correlation between crop yield and various factors is indicated by the connecting lines. * $p < 0.05$, ** $p < 0.01$, *** $p < 0.001$.

## 5. Conclusions

The present results showed that different maize–soybean bandwidth row ratio planting configurations affected nutrient accumulation, interspecific relationships, interspecific dominance, and yield of the crops. Bandwidth of 2.4 m and row ratio of 2:4 showed superior land productivity and intercropping advantage, and in this model, intercropped maize had stronger nutrient acquisition ability, and intercropped crops had higher net effect, select effect, and complementarity effect. Maize–soybean intercropping had positive complementarity and select effect, but the intercropping advantage of this model was mainly based on complementarity effect. With the screening of herbicides and chemical regulators for intercropping crops, mechanization and other problems are further improved, and combined with the appropriate bandwidth row ratio configuration planting pattern, our research could provide an important reference for the future application and promotion of the maize–soybean strip intercropping model in global countries.

**Author Contributions:** Writing—original draft preparation and edition, L.F., W.Y. and H.T.; project administration, S.W.; conceptualization, methodology and investigation, L.F., W.Y., G.H. and S.W. writing—review and editing, L.F. and W.Y. All authors have read and agreed to the published version of the manuscript.

**Funding:** This study was supported by the following funding sources: The National Key Research and Development Program of China (No.2016YFD030020906); The National Natural Science Foundation of China (No.31901125).

**Data Availability Statement:** Not applicable.

**Acknowledgments:** I would like to thank Shubin Wang and Guoqin Huang for the guidance on experimental design and thank Wenting Yang and Haiying Tang for the valuable comments on the manuscript, as well as the comment and suggestions of the anonymous reviewers.

**Conflicts of Interest:** The authors declare no conflict of interest.

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
