# Peer review of "Bandwidth Row Ratio Configuration Affect Interspecific Effects and Land Productivity in Maize–Soybean Intercropping System"

_agronomy, doi:10.3390/agronomy12123095_

Round 1
Reviewer 1 Report
Dear authors,
the experiment showed a confounded effect of inter-cropping and row spacing. A seperation is not possible, thus the experimental design in inadequate to answer your research questions, sorry for saying this. See some additional hints for statistical analysis (note there are more points that need revision). Note that accounting for all of them does not solve the first point mentioned.
The research question adressed by the authors is to check which way of intercropping maize and soybean had advantage on biomass and nitrogen use. Most traits analysed use the comparison with sole-cropping directly by calculating ratios (LER) or indirectly by comparing means and using mean differences in an ANOVA. Unfortuatelly, the experimental design does not allow to answer this question. The authors compared sole cropping with 50/70 cm inter-row spacing with intercropping with 40 cm row-spacing. Intercropping resulted in up to 14% more maize plants and (at the same time!) 67% more soybean plants per area. If there are more plants per area, production is probably higher for biomass and N uptake, this does not prove any advantage of inter-cropping. The design lack treatments that vary row-spacing or inter-cropping/sole-cropping alone.
Beside this, the statistical description lacks information on the model. From results it can be seen that the authors used simple correlations across response variables where a bivariate model is needed. The performed ANOVA probably used homogeneous variances while heterogeneous variance are most often expected between years. There is no information given on the design, in case it is not a CRD the design is not considered in the analysis - but all these comments are useless, as even with a proper analysis the performed experiment cannot prove effects of intercropping (or row-spacing), as both effects are completely confounded by design.
More detailed answers on specific questions are given below.

Author Response
1、The authors performed a two- or three-factorial design. The first factor is intercropping vs sole cropping. He second factor is a two-by-two design of number of soybean rows and spacing between maize and soybean rows resulting in two bandwidths of 2.4 and 2.8 m. Additionally, row spacing was varied between sole und inter-cropping. In sole cropping, row spacing was 70 cm in maize and 50 cm in soybean. In intercropping, row-spacing was 40 cm. To distinguish between effects of row-spacing and sole/intercropping, the same row spacing should be used in both treatments, thus a row-spacing of 70 cm in intercropped maize and a row-spacing of 40 cm in sole-crop maize. Unfortunately, these treatment combinations are missing. The experiment therefore does not allow to distinguish between effects of intercropping and row-spacing.
Response: Thank you for this insightful comment. We used a single factor randomized group design and the trial design was not flawed; other reviewers did not point out such critical errors and they were all in favor of the trial design of this manuscript. Our single-crop planting model is only designed to form a control group with the intercropping model. The maize soybean strip intercropping model has three critical strategies: optimal cultivar screening, in-row plant spacing reduction, and row spacing expansion. We planted two rows of maize at 40 cm spacing in the intercropping pattern in order to increase the light energy interception of four rows of soybean plants, improving the light energy use efficiency of intercropped soybean, and thus increasing the population yield and economic efficiency of the intercropping system. With the traditional maize monocrop row spacing (70 cm) as a control group, we optimized the traditional cropping pattern and ensured that the density of the intercrop and monocrop remained the consistent and reduced the spacing between the intercrop maize and intercrop soybean plants.
- Feng, L., Tang, H. Y., Pu, T., Chen, G. P., Liang, B., Yang, W. Y., Wang, X. C. Maize soybean intercropping: a bibliometric analysis of 30 years of research publication. Agron. J. 2022. https://doi.org/10.1002/agj2.21186.
- Du, J. B, Han, T. F, Gai, J. Y, Yong, T. W, Sun, X., Wang, X. C, Yang, F., Liu, J., Shu, K., Liu, W. G, Yang, W. Y. Maize-soybean strip intercropping: Achieved a balance between high productivity and sustainability. J. Integr. agric. 2018, 17, 747-754. https://doi.org/10.1016/S2095-3119(17)61789-1.
2、Just assume that we have two-times 7 plots (total 14) of B1R2 and 8 3-row plots of sole-crop maize and sole-crop soybean with similar width (width is 16.8 m in all cases). In intercropping there are 2 rows maize per plot resulting in 28 rows in total. In sole cropping maize there are 3 rows per plot resulting in 24 rows. For soybean, we have 4 rows in intercropped plots resulting in 56 rows compared to 33.6 rows in sole-cropping. Therefore, the calculation of LER compared 28 rows maize+56 rows soybean with 24 rows maize and 33.6 rows soybean (+14% maize and +67% soybean!). Is it really surprising that biomass and N accumulation differ, if N is not limited?
Response: Thank you for this insightful comment. We ensure that the densities of intercrop and sole plantings are kept equal, and the spacing between monocrop plants is greater than the spacing between intercrop plants, in addition, and that we have one intercrop treatment corresponding to one single cropping treatment. Then, according to your calculation method, there are two times intercrop treatment plots, i.e. 28 rows of maize + 56 rows of soybeans, which should also be two times the number of maize single-crop plots and two times the number of soybean single-crop plots for the control group, i.e. 48 rows for maize and 67 rows for soybeans. On the other hand, LER shows the yield advantage of the intercropping crops, which is generally calculated in terms of yield and not in terms of number of intercropped rows, etc. Our LER calculation is based on crop yield, and according to the previous calculation. Land equivalent ratio (LER): as a measure of yield advantage.
LERM = Yim/Ysm, LERS = Yis/Yss, LER = LERM + LERS
Where LERM and LERS are the partial land equivalent ratio of maize and soybean, respectively. Yim and Yis are the intercropped maize and intercropped soybean yields, respectively. Ysm and Yss are the sole maize and sole soybean yields, respectively. When LER > 1, it shows intercropping advantage, and when LER < 1, it shows intercropping disadvantage.
All of our previous trials ensured equal fertilizer application rates for individual maize plants in both intercrop and monocrop cropping patterns. We ensured standard intercrop and monocrop maize fertilizer application rates and fertilizer application rates were strictly required. The nitrogen fixation by the root nodules of intercropped soybeans can reduce or without the application of nitrogen fertilizer, which can effectively improve fertilizer efficiency.
3、From my point of view, the current experimental design does not allow to draw conclusions about intercropping/sole-cropping alone. The current experiments allow to compare the four intercropping treatments, but calculation LER or other ratios with sole-cropping does not make sense, as the sole-crop/intercropping effects is completely confounded with the row-spacing effect. Therefore, the design is inadequate to answer large parts of the research questions.
Response: Thank you for this insightful comment. Our experimental design has been critically discussed by several experts and there are no issues with the trial design, and relevant papers have been published. We now conclude that intercropping is superior to traditional monocropping (LER>1) and has a better yield advantage than single-crop maize or single-crop soybean with LER=1. B1R2 has the best productivity and can increase crop yield and farmers' income.
4、The authors performed the experiment at a single location. No prediction for other locations are possible from this type of data.
Response: Thank you for this insightful comment and precious advice. Our trial was conducted in Jiangxi Province, China, of course, different geographical locations make different light, heat and water resources and soil environment, which will cause yield differences under intercropping systems, but through our preliminary research results, it is clear that the maize soybean bandwidth row ratio configuration model has yield advantages, moreover, with 106 ha of maize soybean strip planting area in China in 2022, our trial is also very meaningful and valuable, and can provide reference programs for other regions. With the expansion of our maize soybean strip planting area, other regions have accordingly screened out suitable planting patterns with bandwidth row ratio configurations, which provide technical support for increasing
5、The authors used an LSD, which is valid in case the global F test is significant. The authors used the LSD test for both main effects and interaction effects at the same time. If there are interaction effects, an LSD test for main effects does not make sense, otherwise the test for interaction effects should be dropped.
Response: Thank you for this insightful comment and precious advice. We accept your suggestion and we have deleted the analysis of interaction effects. Please see the article.
6、The linear regressions fitted in Figure 1 are senseless. There is no degree of freedom to test the regression. The shown dashes simply connect mean values, and where probably not estimated at all. Dropping the dashes is the easies way to solve the issue.
Response: Thank you for this insightful comment and precious advice. Figure 1 shows the precipitation and average temperature during the experiment, and no linear regression was done.
7、The explanation of letter display is wrong. Means with at least one identical letter are nonsignificant from each other, which is not the same as means with different letters show significant differences. As XAs=XAm figure 5 can be reduced to either positive or negative values.
Response: Thank you for this insightful comment and precious advice. The positive values are the aggressivity of intercropping maize and the negative values represent the aggressivity of intercropping soybean, we also refer to the published articles, because the positive or negative values represent the competition level of different intercrops, if the positive and negative values are shown in one graph, we can directly see the competition level of intercropping system, which is beneficial to the optimization and improvement of our intercropping system.

Reviewer 2 Report
The manuscript concerns the combined cultivation of soybean and maize intercroping system in China. Although the research is of great cognitive and scientific importance, as it examines and documents the effects of the proximity of one plant to another, I am skeptical about the possibility of using such cultivation on a large scale under the conditions of using equipment for sowing and harvesting with a working width equal to a multiple of the tested distances between rows of maize and soybean.
Author Response
Reviewer 2:
The manuscript concerns the combined cultivation of soybean and maize intercropping system in China. Although the research is of great cognitive and scientific importance, as it examines and documents the effects of the proximity of one plant to another, I am skeptical about the possibility of using such cultivation on a large scale under the conditions of using equipment for sowing and harvesting with a working width equal to a multiple of the tested distances between rows of maize and soybean。
Response: Thank you for this insightful comment. Our research team has cooperated with machinery team experts to develop sowing and harvesting machines suitable for maize soybean strip planting. We can produce appropriate soybean harvesting machines according to different soybean row distances (Figure D), and mini machines with 2.0 and 2.4 meters bandwidth are getting much attention now, and these machines have been produced and applied, which greatly improve the efficiency. Our team, with the support of the Chinese government, has planted 15 million acres of maize soybean strip planting in 2022, which significantly improves crop yield, and we are expecting to sow 20 million acres in 2023. All of our maize soybean planting uses mechanical sowing and mechanical harvesting, and the problems of mechanical sowing and harvesting have been initially solved. Thank you again for your professional comments.

Reviewer 3 Report
The article presents interesting scientific results. The paper is well prepared. In agricultural sciences, however, the possibility of direct use of the results in agricultural practice or limitation in such use should be indicated. This is missing from the article and should be supplemented.
Several detailed comments and suggestions for improving the manuscript are included in the text as comments.

Author Response
1、Intercropping plays an indispensable role in sustainable agriculture. Explain why?
Response: Thank you for this insightful comment. Maize and soybean, as the typical representatives of the Gramineae and Leguminosae, are called as the best partners in intercropping system. Soybean provides the nitrogen to the maize growth through nitrogen fixation from the air, and the amount of fixed nitrogen transported to maize is 25-155 kg/hm2. The main mechanism is that most of the nitrogen fixed by soybean rhizobia is provided to maize for uptake and utilization, and the large demand for nitrogen by maize is supplied. Due to the increased uptake of nitrogen by maize, the content of nitrogen in the soil is reduced, thus alleviating the inhibitory effect of high nitrogen content on the activity of nitrogen-fixing enzymes, increasing the amount and efficiency of nitrogen fixation by soybean rhizobia, and alleviating the disadvantaged position of soybean in the intercropping system. Gramineae can promote the secretion of inter-root flavonoids, etc., and effectively enhance the nodulation and nitrogen fixation of legumes. The legumes can better fix N2 in the air due to their own root nodule nitrogen fixation, which can provide better nitrogen supply for adjacent crops and also reduce nitrogen application. It supports the reduction of greenhouse gas emissions, improving soil texture and sustainable development in China.
2、what are the difficulties and limitations of this method of growing plants in practical agriculture?
Response: Thank you for this insightful comment. Maize soybean intercropping has had a history of cultivation in China for many years, and the planted area has been increasing year by year, playing a major role in China's food security. Although maize soybean strip planting technology has been widely promoted, there are still many issues, such as: 1. selection of field herbicides, because maize and soybean belong to different kinds of crops, so there are certain differences in the selection of field herbicides, if soybean herbicides migrate to the maize crop, it will lead to the death of intercropped maize. 2. mechanized harvesting problems, maize soybean strip planting technology can achieve small-scale mechanized harvesting in southwest China, but mechanical research suitable for other regions needs to be further improved, with the eventual solution of mechanical harvesting problems, however, our work will be further recognized by others.
3、soil systematics according to WRB and soil texture.
Response: Thank you for this insightful comment and precious advice. We accept your suggestion and we have revised the manuscript and hope that our version will be approved.
4、month name
Response: Thank you for this insightful comment and precious advice. We accept your suggestion and we have revised the manuscript and hope that our version will be approved.
5、what basis these varieties were selected.
Response: Thank you for this insightful comment and precious advice. We accept your suggestion and we have revised the manuscript and hope that our version will be approved.
6、please present as references item
Response: Thank you for this insightful comment and precious advice. We accept your suggestion and we have revised the manuscript and hope that our version will be approved.
7、Discussion and conclusion, there should be a reference to the possibilities and limitations of using the results of these studies in practical intensive agriculture.
Response: Thank you for this insightful comment and precious advice. We accept your suggestion and we have revised the manuscript and hope that our version will be approved.

Round 2
Reviewer 1 Report
please see the attachment

Author Response
Accompanying letter
Dear editors and reviewers:
Thank you for your letter and for the reviewer's comments concerning our manuscript entitled “Bandwidth row ratio configuration affect interspecific effects and land productivity in maize soybean intercropping system (agronomy-2029100)”. Those comments are all very valuable and very helpful for revising and improving our paper, as well as the important guiding significance to our research. We have studied comments carefully and have made the correction which we hope meet with approval. The main correction in the paper and the responses to the reviewer's comments are as flows:
Responds to the reviewer's comments:
Reviewer 1:
1、The authors analysed a two-year single-location experiment with six different treatments. A single location experiment does not allow to make prediction for areas outside the location considered, as treatment effects and treatment-by-location effects are confounded. Moreover, year effects were probably taken as fixed (no information is given about this, but means were presented in the result section), thus estimation is limited at the two years considered. For me there is little gain from knowing the yield of treatments in Jinxing experimental station four or three years ago.
Response: Thank you for this insightful comment and precious advice. It is true that we chose only one location for our experiment to study mainly treatment effects and inter-year effects. Our results showed that the 2.4 m bandwidth, 2-row maize intercropped with 4-row soybean model was better in terms of group yield and nutrient utilization, and the tendency was consistent for 2 years. We did not discuss the interannual effect because the trends of the results were more consistent in both years. In addition, our planting model design has been guided by many experts and is a result of a combination of multiple trial sites in China, and this similar model has been promoted to 15 million hectares across China in 2022, and is expected to increase to 20 million hectares in the next few years. Once again, thank you for your valuable suggestions, and we will consider setting up more trial sites to verify the model according to the climate characteristics in the subsequent trials for better application.
2、The authors used a constant seed density in their treatments. Using the information presented in the manuscript, they used in sole cropping maize 60 k kernels per ha, which fits to a row spacing of 0.7 m and a inter-row distance of 23.8 cm (1/0.238*1/0.7*10000=60000). In treatment B1R1, the authors used two maize rows per band-width and have a inter-row distance of 13.8 cm. This resulted in 1/0.138*2/2.4*10000=60000 kernels per ha. But note that the authors were additionally growing soybeans on the same area with another 150000 kernels on the same area. Thus the authors compared sole cropping with intercropping, but doubled the number of kernels sown. The number of used seeds in one ha sole crop maize plus one ha sole crop soybean is equal to the number of seeds used in one ha intercropping. In this case, it is no surprise that inter-cropping showed advantages in biomass and so on.
Response: Thank you for this insightful comment and precious advice. This is true. In order to ensure the yield of the main crop maize, our maize soybean strip intercropping planting pattern is characterized by ensuring the same planting density and single plant fertilizer application between intercropping and monocropping by reducing the distance between plants of intercropping crops to ensure the same density as monocropping, such intercropping pattern has been carried out in Sichuan province and Shandong province in China, and this planting pattern makes full use of light, temperature, water and other natural resources and can significantly improve land productivity. At the same time, our maize soybean strip intercropping planting is also conducive to reducing the application of chemical fertilizers, reducing inputs, improving soil quality, and improving resource use efficiency, which is a green, ecological and sustainable cropping pattern.
3、As stated before, the authors fixed two effects in their study, first sole- and intercropping and second seeds per ha. A seperation of both is not possible. Note that six maize plants per m² in sole-cropping are compared to 14.5 plants in B1R1. To seperate both effects, sole- and inter-cropping should be tested keep seed density constant, and seed density should be tested keeping the cropping system constant. These treatment combinations are missing.
Response: Thank you for this insightful comment and precious advice. I would like to explain that we established the selection effect and compensation effect of the intercropping model in our experiment and did not intend to distinguish between monocrop and intercrop effects, per hectare seed effect. We wanted to evaluate the crop yield advantage under different bandwidth row ratio configuration patterns using the ecological theory of selection and compensation effects to elucidate the relationship between selection and compensation effects on crop yield. Our objective was to screen a maize soybean strip intercropping planting pattern with harmonious intercrop relationships and better population yields, thus setting up an intercrop pattern and a monocrop pattern treatment. We chose monocropping to show that the maize soybean strip intercropping planting pattern is better than the traditional monocropping pattern, and we used monocropping maize and monocropping soybean to compare with each intercrop treatment.
We ensure that the planting density is consistent when planting, if the spacing between intercrop and monocrop is consistent, then the yield of intercrop maize will be greatly reduced, which does not embody the advantages of our maize soybean strip intercropping planting pattern, and will not improve the efficiency of land utilization. Our group has proven over 20 years of experiments that reducing the plant spacing of intercropping crops, combined with a suitable row ratio configuration planting pattern, while ensuring the same planting density of monocropping and intercropping crops, can achieve no reduction in intercropped maize yield and an additional 1500-1800 kg/ha of intercropped soybeans, which is conducive to improving crop yield [1-7].
[1] Optimum leaf defoliation: A new agronomic approach for increasing nutrient uptake and land equivalent ratio of maize soybean relay intercropping system - ScienceDirect[J]. Field Crops Research, 244:107647-107647.
[2] Liu W, Deng Y, Hussain S, et al. Relationship between cellulose accumulation and lodging resistance in the stem of relay intercropped soybean [Glycine max (L.) Merr.] [J]. Field Crops Research, 2016.
[3] Liu X, Rahman T, Song C, et al. Changes in light environment, morphology, growth and yield of soybean in maize-soybean intercropping systems[J]. Field Crops Research, 2017, 200:38-46.
[4] Yang F, Liao D, Wu X, et al. Effect of aboveground and belowground interactions on the intercrop yields in maize-soybean relay intercropping systems[J]. Field Crops Research, 2017.
[5] Chen P, Song C, Liu X M, et al. Yield advantage and nitrogen fate in an additive maize-soybean relay intercropping system[J]. The Science of the Total Environment, 2019, 657(MAR.20):987-999.
[6] DU J, HAN T, GAI J, et al. Maize-soybean strip intercropping: Achieved a balance between high productivity and sustainability[J]. Journal of integrative agriculture, 2018, 17(4): 747-754.
[7] Iqbal N, Hussain S, Ahmed Z, et al. Comparative analysis of maize–soybean strip intercropping systems: a review[J]. Plant Production Science, 2019, 22(2): 131-142.
……
4、The authors throughout the manuscript claimed that they showed that inter-cropping outperformed sole-cropping. But the effect can partly or completely be based on the differences in seed density.
Response: Thank you for this insightful comment and precious advice. Indeed, a large part of this intercropping advantage is due to the guaranteed planting density of the intercrop. It was demonstrated through trials that, despite the reduced spacing of plants, maize soybean strip intercropping planting pattern can take full advantage of light energy because the two crops belong to different crops, with taller maize receiving higher ecological niches of light energy and shorter soybeans receiving lower ecological niches of light energy, allowing light energy resources to be fully utilized. At the same time, legume crops can provide nitrogen nutrients to maize, and maize roots can obtain nitrogen from soybean roots, which facilitates better crop yields for intercropped maize, while maize soybean intercropping is also known as the best intercropping partner. Growing two crops on the same plot of land at the same time also helps to increase the land productivity of the crop. This is the core of our maize soybean strip intercropping planting pattern.
5、Beside this serious flaw in designing the experiment, there are several error in the statistical analysis and description of material and methods.
Response: Thank you for this insightful comment and precious advice. Our experimental design has been discussed by many experts before implementation, and we are sorry for any confusion caused by some contents we may not have written clearly enough for you to read. We have carefully revised the method section, please see the text.
6、The authors claimed that they are using a completely randomized block design. There are two experimental designs denoted as completely randomized design (CRD) and a randomized complete block design (RCBD), a completely randomized block design is a contradiction in itself, as a block design always restrict randomization to block and thus cannot be completely randomized.
Response: Thank you for this insightful comment and precious advice. We have carefully revised the method section, please see the text.
7、The authors used the term very significantly e.g. in the abstract. A result is statistically significant, by the standards of the study, when p ≤ α. There is no very significant as the test decision is binomial (non-significant or significant). It seems to be that the authors believe that very significant or a smaller p-value make results more relevant. For relevance, the authors have to consider effect sizes.
Response: Thank you for this insightful comment and precious advice. We have carefully revised the method section, please see the text.
8、In equations 1 to 8 it is unclear what the authors did. From seeing the equation (variable do not have indices), the calculation is done once. In contrast, I argue that the authors did this per plot or another unit in the experiment. Please clarify and correct equations.
Response: Thank you for this insightful comment and precious advice. We have carefully revised the method section, please see the text.
Nutrient accumulation
Nitrogen accumulation (NA) = weight of dry matter (kg ha−1) × N (%)/100. |
(1) |
Crop phosphorus accumulation (PA), potassium accumulation (KA) was calculated consistent with NA.
Intercropping advantage
(1) Land equivalent ratio (LER): as a measure of yield advantage.
LERM = Yim/Ysm, LERS = Yis/Yss, LER = LERM + LERS |
(2) |
Where LERM and LERS are the partial land equivalent ratio of intercropped maize and soybean, respectively. Yim and Yis are the intercropped maize and intercropped soybean grain yields, respectively. Ysm and Yss are the sole maize and sole soybean grain yields, respectively. When LER > 1, it shows intercropping advantage, and when LER < 1, it shows intercropping disadvantage.
(2) Nitrogen equivalent ratio (NER): As a measure of nitrogen uptake advantage in crops.
NERM = Nim/Nsm, NERS = Nis/Nss, NER = NERM + NERS |
(3) |
Where NERM and NERS are the partial nitrogen equivalent ratio of maize and soybean, respectively. Nim and Nis represent nitrogen accumulation per unit area in aboveground intercropped maize and intercropped soybean, respectively. Nsm and Nss represent nitrogen accumulation per unit area in aboveground sole maize and sole soybean, respectively. When NER > 1, it shows intercropping advantage, and when NER < 1, it shows intercropping disadvantage. The phosphorus equivalent ratio (PER) and potassium equivalent ratio (KER) are calculated in the same way as the NER.
(3) Net effect (NE), complementarity effect (CE) and select effect (SE): The NE includes two components: SE and CE.
(4) |
|
(5) |
|
(6) |
NE > 0, indicating the positive effect between diversity planting and intercropping system productivity, NE < 0, indicating negative effect. SE > 0, indicates that high yielding species will be increased in the intercropping system, SE < 0, low yielding species will be increased in the intercropping. Pm and Ps represents the percentage of plot area occupied by intercropped maize and intercropped soybean, respectively. In our study, the plots were occupied by maize as Pm = (maize-maize row spacing + maize-soybean row spacing)/bandwidth, for example, the area occupied by maize in plot B1R1 was Pm = (40 cm + 60 cm)/240 cm = 0.42, the area occupied by maize in the other plots was 0.33, 0.43, 0.36, and the area occupied by soybeans in the plots was 0.58, 0.67, 0.57, 0.64. Yim and Yis are the intercropped maize and intercropped soybean yields, respectively. Ysm and Yss are the sole maize and sole soybean yields, respectively.
(4) Nutrient Aggressivity (NA): indicates the relative nitrogen uptake of intercrop A compared to the advantage of nitrogen uptake of intercrop B.
NAM = Nim/Nsm × Pm − Nis/Nss × Ps |
(7) |
NAS = Nis/Nss × Ps − Nim/Nsm × Pm |
(8) |
NAM and NAS are represented as intercropped maize nitrogen aggressivity and intercropped soybean nitrogen aggressivity, respectively. NAM = NAS, indicated that maize and soybean had the same competitive advantage for nitrogen uptake; NAM >NAS, means that intercropped maize has a stronger nitrogen uptake advantage than intercropped soybean, NAM < NAS, means that intercropped maize has less advantage in nitrogen uptake than intercropped soybean. The phosphorus aggressivity (PA) and potassium aggressivity (KA) are calculated in the same way as the NA.
9、The authors denied to specify their model assumptions. They used an ANOVA approach, but no model is given, so there is no information about what the authors did.
Response: Thank you for this insightful comment and precious advice. We performed post hoc multiple comparisons and Duncan's test, and based on the common results of these two tests, we indicated the significance of each treatment by means of the letters a and b. The results of the multiple comparisons were consistent with the results of Duncan's test, so we did not provide information on the comparison of multiple means in order to prevent data redundancy. The means and standard errors were written based on the results of the descriptive statistics, and we made graphs based on the results, so that the values of the various treatments, the values of the multiple comparisons, and the significance analysis are found in the figures.
10、Description of letter display is still wrong. Note that two means with letters ab and b care not significant different while the authors description indicated significance.
Response: Thank you for this insightful comment and precious advice. We have carefully revised the method section, please see the text.
11、The authors used Pearson correlations between response variables (they call factor of intercrop advantage, but traits are not a factor). As observations are modelled as a sum of effects (not sure how the sum looks like, as the model is not specified), the correct method is a bivariate analysis.
Response: Thank you for this insightful comment and precious advice. We have performed tests in the process of writing our manuscript. The results of bivariate correlation analysis and Pearson correlation analysis are the same, and we have used graphing software to analyze the correlation among the factors and Mantel method to analyze the correlation between each factor and crop yield, which makes our manuscript more beautiful.
12、In Figure 3 the authors presented treatment-by-year means and treatment means. The former should be presented in case of treatment-by-year interaction, the latter in case of obsense of interactions. Unfortunatelly, results from ANOVA were not given. From figure 3/4 it seems to be that the authors used treatment-by-year-specific variances. No justification for this model assumption was given. From description ordinary means were presented. Using least square means from the modelled fitted is the state of the art method that should be used. Note that ordinary means and weighted least square means are identical in case of balanced data. For unbalanced data, ordinary means are wrong.
Response: Thank you for this insightful comment and precious advice. We give the results of the analysis of variance in Figure 3, because the results of the Duncan's test are consistent with the results of the post hoc multiple comparisons, so we applied the results of Duncan's test, with letters indicating the significance results. We averaged the data during the experimental period, which is a basic treatment for multi-year trials, because climatic factors such as climate and temperature affect the experimental results each year, so we averaged the results for two years, and on the other hand from the average we can better see where the differences among treatments lie.
13、Figure 5 still presented identical upper and lower bars. Identity of bars is given by the definition. One bar would be enough. Headers in Table 2 are miss-leading as / cm does not mean per cm but in cm. The slash means divided by. Sentences should start with an upper-case letter, e.g. lowercase… in the description of figures.
Response: Thank you for this insightful comment and precious advice. The error bars at the upper part of Figure 3 represent the standard error of intercropped maize, while the error bars at the lower part represent the standard error of intercropped soybean, which are two kinds of different meanings, so we keep both error bars.
Thank you again for your precious suggestions and opinions on our manuscript, and thank you for reviewing our manuscript, which is to further improve its quality and make it more acceptable to readers. We hope that our revisions will be recognized by you and that our manuscript would be accepted. Once again, thank you.
Round 3
Reviewer 1 Report
Please carefully revise comments given earlier. Even typing errors were not corrected.
Author Response
Dear Reviewer,
The manuscript sentences in capital letters represent the abbreviated form. thanks, in addition, our experimental design has been discussed by many experts before implementation. Thank you for this insightful comment and precious advice.
